# Breaking the Mold: Epigenetics and Genomics Approaches Addressing Novel Treatments and Chemoresponse in TGCT Patients

**DOI:** 10.3390/ijms24097873

**Published:** 2023-04-26

**Authors:** Berenice Cuevas-Estrada, Michel Montalvo-Casimiro, Paulina Munguia-Garza, Juan Alberto Ríos-Rodríguez, Rodrigo González-Barrios, Luis A. Herrera

**Affiliations:** 1Unidad de Investigación Biomédica en Cáncer, Instituto Nacional de Cancerología-Instituto de Investigaciones Biomédicas, UNAM, Mexico City 14080, Mexico; 2Tecnológico de Monterrey, Escuela de Medicina y Ciencias de la Salud, Monterrey 64710, Mexico

**Keywords:** TGCT, cisplatin, chemoresponse, epidrugs, sensibility, genomics and epigenomics

## Abstract

Testicular germ-cell tumors (TGCT) have been widely recognized for their outstanding survival rates, commonly attributed to their high sensitivity to cisplatin-based therapies. Despite this, a subset of patients develops cisplatin resistance, for whom additional therapeutic options are unsuccessful, and ~20% of them will die from disease progression at an early age. Several efforts have been made trying to find the molecular bases of cisplatin resistance. However, this phenomenon is still not fully understood, which has limited the development of efficient biomarkers and precision medicine approaches as an alternative that could improve the clinical outcomes of these patients. With the aim of providing an integrative landscape, we review the most recent genomic and epigenomic features attributed to chemoresponse in TGCT patients, highlighting how we can seek to combat cisplatin resistance through the same mechanisms by which TGCTs are particularly hypersensitive to therapy. In this regard, we explore ongoing treatment directions for resistant TGCT and novel targets to guide future clinical trials. Through our exploration of recent findings, we conclude that epidrugs are promising treatments that could help to restore cisplatin sensitivity in resistant tumors, shedding light on potential avenues for better prognosis for the benefit of the patients.

## 1. Introduction

Testicular cancer (TCa) is a relatively rare neoplasm on a per-population basis; it accounts for 1 to 2% of all neoplasms in men, with a GLOBOCAN estimate of 74,458 new cases and 9334 deaths in 2020 [1]. TCa incidence has been rising over the past 35 years, and it is expected to continue this way, especially in highly populated countries, due to their population structure [2]. Racial differences have always been noted, with rates among young white men being >10 times higher than those observed in black or Asian mean. However, there is a rising tendency among all ethnicities [3].

Testicular germ cell tumors (TGCT) comprise 95% of TCa cases, but their etiology is not fully understood. Nevertheless, the fact that their prevalence varies across different populations implies that environmental influences or genetic factors might be involved. In fact, the best-characterized risk factors for TGCT include cryptorchidism, hypospadias, infertility, gonadal dysgenesis, testicular microlithiasis, family history of TGCT, previous diagnosis of TCa, and exposure to environmental factors [4].

Over time, TGCTs have been recognized as having a therapeutic model due to their outstanding sensitivity to cisplatin-based treatment, which has yet to be replicated in other types of cancer. Survival rates have increased from less than 30% in the 1950s to approximately 95% today [5]. Even patients with disseminated TCa achieve up to 80% cure rates after first-line chemotherapy. However, refractory disease is observed in about 20–30% of patients, and the second line of salvage chemotherapy only results in an overall response rate (ORR) of 20%, leaving limited therapeutic options for these remaining cases. Approximately 5% of all TGCT patients ultimately succumb to their disease due to cisplatin resistance [6]. To date, the molecular mechanisms underlying cisplatin resistance have been elusive, which makes it difficult to discover new biomarkers for predicting response to therapy and the development of targeted therapies that may function as clinical alternatives in these patients [7].

In this review, we summarize current genomic and epigenomic features associated with chemoresponse in TGCT patients and present an overview of the independent molecular profiles of sensitive and resistant tumors. Additionally, we discuss recent advances in research on new therapies, with special attention to epidrugs as a promising alternative to enhance cisplatin response by resensitizing resistant tumors.

## 2. Summary of TGCT’s Major Features

### 2.1. Histological TGCT Subtypes

TGCTs are classified into two main histological categories: seminoma (SE) and non-seminoma (NS). Both derive from a preinvasive lesion known as a germ cell neoplasm in situ (GCNIS), composed of primordial germ cells arrested in their maturation process into gonocytes [6]. These cells express pluripotency markers, do not differentiate, and remain dormant until puberty when they gradually accumulate chromosomal abnormalities that activate malignant growth and allow progression to SE or embryonal carcinoma (EC). These cells maintain pluripotent properties, and EC can give rise to the rest of the tumor subtypes of the NS category: yolk sac tumors, choriocarcinoma, and teratoma [8].

SEs account for approximately 50% of all testicular tumors; however, they are the most common histological subtype of TGCT in men aged under 45. This TGCT subtype is particularly susceptible to chemo- and radiotherapy agents and usually displays a good prognosis [9]. Recently, a computational analysis of omics data on 64 pure seminoma samples from the TCGA database revealed the existence of two distinct seminoma subtypes. Subtype 1 has a higher pluripotency rate, and subtype 2 exhibits the characteristics of a more differentiated cell type and homologous recombination (HR) repair deficiency, suggesting that this subtype could be responsible for the few cases of recurrence in SE patients after chemotherapy [10].

NSs are characterized by a variety of differentiation patterns from both embryonic and extraembryonic tissues, and they are often mixed tumors. NSs have more aggressive clinical behavior, requiring more intensive treatment approaches [4]. Mature teratomas showing the highest degrees of differentiation are recognized as the most chemotherapy-resistant TGCT cases [11,12,13]. 

Understanding the cellular origin and distinctive features of these tumors could help clarify their behavior. Here, we establish the first clue to address the phenomenon of chemoresistance in TGCT; it seems that there is a direct relationship between the differentiation state of the cells and chemoresponse, and the degree of pluripotency correlates with chemosensitivity [14].

### 2.2. TGCT Genomic Hallmarks

As previously established, TGCTs are histologically complex, with various degrees of differentiation and pluripotency playing a major role in heterogeneity. This, together with complex oncogenesis, rare actionable mutations, unpredictable clonal evolution, and molecular heterogeneity has made research and development on biomarkers a clinical challenge [15]. Diverse whole-exome sequencing (WES) studies have revealed some key features for understanding the genomics underlying TGCTs, discussed as follows.

#### 2.2.1. Actionable Mutations

In general, TGCTs, both SE and NS, have a low tumor mutation burden (TMB) (0.5 mutations per Mb) [16,17,18,19], in contrast to other malignancies such as melanoma (14.4/Mb) [20] or colorectal cancer (9.9–11.6/Mb) [21]. The low frequency of mutations aligns with the postulated embryonic source of TGCT [17] and has limited the identification of actionable somatic mutations for diagnostic biomarkers. Even worse, the mutations described do not seem to be consistent between studies, possibly because both SE and NS have high intratumoral heterogeneity [16,17,18,19].

Even though mutations in single genes are not common, they still play an imperative role in the development of TGCT. Currently, only three genes with recurrent somatic variants have been identified: *KIT*, *KRAS,* and *NRAS,* all of which are seen exclusively in tumors with seminomatous components (either pure SE or mixed NS) [17,18,19].

The *KIT* (or c-*KIT*) gene is the most strongly associated with TCa, with mutations in 18–30% of all patients. This proto-oncogene encodes for tyrosine kinase protein receptor (RTK) and plays crucial roles in cell survival, proliferation, and apoptosis [22]. A dysregulated *KIT* function, due to either overexpression or mutations, promotes tumor development and progression in various human cancers, with mutations in TGCT being the most well characterized for SE diagnosis and prognosis [22,23]. Despite being a specific mutation of SE, its low frequency does not allow it to be positioned as a universal driver for all SE [24]. *KRAS* and *NRAS* mutations, which also activate downstream pathways related to cell survival, are less frequent than *KIT* mutations and may be associated with more aggressive phenotypes, although their role in the development of TGCT and clinical outcomes is still unclear. Recently, the importance of alternative splicing in the regulation of *KIT, KRAS*, and *NRAS* transcriptional regulators has also been highlighted [25].

The tumor suppressor gene *CDC27* and the testis-specific expression gene *FSIP2* have been associated with the development of TGCTs [26,27]. Other somatic variants have also been reported in genes such as *FGFR3, MDM2, BRAF, AKT1, PIK3CA, RPL5, RAC1*, and *NCOA3* [16,18,28,29]. Nevertheless, since mutations are only present in a minority of patients, a single universal mutational factor cannot explain the development of TGCTs.

Instead, a polygenic nature for testicular cancer has been proposed, with over 80 risk loci reported to date; the accumulation of low-frequency susceptibility genes seems to produce an increased risk for TGCT development. However, the prevalences of the same germline variants have not been comparable through different study cohorts [30,31,32].

#### 2.2.2. Chromosomal Aberrations

In contrast with their low TMB, TGCTs are highly aneuploid and often show large-scale copy number gains and losses [17]. Aberrations in the development of TGCT include gene deletions, chromosomal duplications, and loss of heterozygosity. Overall, TGCT are markedly aneuploid, with recurring chromosome gains in 2p, 7, 8, 12, 14q, 15q, 17q, 21q, and X, as well as deletions on chromosomes Y, 4, 5, 11q, 13q, and 18q2, which are consistently reported at a frequency of 25–40% [16,18,19,33]. In general, the currently known chromosomal aberrations have been more commonly described in less aggressive tumor subtypes [16,18]. WES and genome-wide sequencing (WGS) analysis indicate that chromosome aberrations are consistent in most patients regardless of population differences in the cohorts studied [18,18,19,33]. While NS are typically hypotriploid, SE are hypertriploid. In addition, the most common abnormality is the presence of a 12p isochromosome (i(12p)), which affects more than 80% of TGCTs [17,19]. The *KRAS* proto-oncogene and various stem-cell-associated genes, including *NANOG* and *STELLAR*, are situated within the minor arm of chromosome 12. Notably, i(12p) is not apparent in pre-cancerous growths [34], which suggests that this genetic abnormality is not an early-phase event in the development of TGCT. Amplification of the *KIT* gene at 4q12 was observed in 21% of SEs and 9% of NSs, leading to high levels of protein expression [29,35].

#### 2.2.3. Potential Genomic Biomarkers for Diagnosis

Currently, there are few potential genomic biomarkers for the diagnosis and clinical prognosis of TGCTs, due to their low mutation rate and the difficulty of transferring the detection of recurrent chromosomal aberrations to the clinic. Although definitive diagnosis is only made after surgical resection on histopathological assessment, the karyotype for the identification of i(12p) is the most accepted genomic-related biomarker for the diagnosis of TGCTs. This is followed by the overexpression of *KIT* and the identification of three different variants for the diagnosis of SE (D816V (A/T); D816H (G/C); N822K (A/T)), which are also related to cryptorchidism [36].

Overall, the assessment of diagnostic biomarkers remains complex. In fact, all the potential genomic biomarkers are in early preclinical research because, all too often, assays are not finally validated in clinical assays, engendering misleading assumptions about biomarker value [37]. However, TGCT clinical management could benefit from research on sensitive biomarkers to guide decision-making [38], including on prognostic outcomes through evaluation of the expression levels of some key players, such as tissue-level overexpression of *OCT4*, *NANOG*, *MCL1,* and the DNA repair enzyme poly ADP-ribose polymerase (PARP) [29,39,40]. Furthermore, subexpressions of *ERCC1, XPA,* and *XPF* are common in TGCT in contrast with normal tissue, as well as subexpression of *MDM2,* which is well described as a clinical biomarker of EC. *ALDH1A3* was significantly overexpressed in all histological subtypes of TGCTs compared to normal testicular tissue. Moreover, high levels of *ALDH1A3* and increased *ALDH* activity were found in cisplatin-resistant EC cell lines [41]. Finally, some factors include differing gene signatures for each subtype, whereby SE exhibits overexpression of genes associated with spermatogenesis (*PRAME, MAGEA4*, and *SPAG1*) and NS displays an overexpression of regulatory genes, including *DNMT3B* and *SOX2* [17]. Furthermore, the study of some miRNAs, such as miR-371a-3p and miR-375, with diagnostic and predictive potential in TGCTs (mentioned below) has grown exponentially in recent years, promising genomic biomarkers with high sensitivity and reproducibility [38].

Advances in describing the genomic characteristics of TGCT are focused on decoding the developments of these tumors as well as identifying risk biomarkers, but there have been few efforts to elucidate and predict the genomic causes of resistance to cisplatin-based chemotherapy, which could help in the clinical management of these patients.

## 3. Chemoresponse as a Clinically Unresolved Problem

Cisplatin-refractory disease is observed in about 20–30% of all patients [6]; it has been defined as patients who continually progress under chemotherapy, who relapse/progress after second-line cisplatin-based chemotherapy, or who progress within one month of completing cisplatin-based chemotherapy [42,43].

TGCTs are recognized worldwide for their excellent responses to cisplatin-based therapy, since even patients with advanced disease who don’t show improvements with conventional first-line therapy can be rescued by salvage chemotherapy [6]. However, this approach has important short- and long-term side effects, like infertility, ototoxicity, renal function impairment, cardiovascular disease, hypogonadism, chronic neurotoxicity, and second malignancy development [44,45]. All of this is aggravated by the fact that most of the patients are young men under 45 years of age, and, still, about 3–5% of all TGCT patients fail to respond to established cisplatin-based standard treatments and potentially die of the disease [46].

In 2020, the highest mortality rates were in Central and South America (0.84 and 0.54 per 100,000 respectively), followed by Western and Southern Africa. The lowest were in Northern Europe (0.16 per 100,000) [2]. This is an interesting observation, considering that the latter are the countries with the greatest incidence. The role of environmental factors has been suggested as a possible explanation [6], but it could be related to differences in genetic susceptibility or, in general, to patterns of chemoresistance at a population level [16].

Intrinsic resistance refers to the lack of an objective clinical response after initial treatment and is mainly due to reduced susceptibility to cell death through an established genomic and epigenomic somatic landscape, not to altered DNA damage induction or repair [47,48]. In acquired resistance, the lack of tumor regression occurs after an initial response. Although it is believed that each type of resistance operates through distinct signaling pathways [47,48,49], this is not yet understood, possibly because scientific reports rarely remark on the specific mechanism studied.

Chemoresistance is a matter of great importance, and the disparities between populations are proof of its multifactorial nature, which requires more research to adequately address it. We highlight the need to develop specific and sensitive biomarkers to establish accurate prognosis and predict beforehand response to treatment in TGCT. First, we must identify those patients intrinsically resistant to cisplatin and that require exploration of other targeted therapies that are effective and safe. Secondly, biomarkers can also identify TGCT patients who are hypersensitive to cisplatin so that clinicians can avoid overtreatment and short- and long-term adverse effects.

## 4. Molecular Basis of Chemoresponse

Integrating the molecular features of TGCT could provide valuable insights into the mechanisms underlying chemoresponse as well as identifying potential predictive biomarkers to improve treatment strategies. In this section, we review current knowledge of chemoresponse from both genetic and epigenetic perspectives and provide an overview of the molecular basis of cisplatin resistance and sensitivity in TGCT. Table 1 summarizes identified genomic markers of chemoresponse, while Figure 1 provides the integrative landscape of proposed chemoresponse mechanisms.

Despite the somatic mutation rate being low regardless of the therapy response phenotype, recent studies using cisplatin-resistant cell lines and cisplatin-resistant tumors found that the resistant phenotype had a higher TMB (although not significantly) and more single-nucleotide variants (SNPs) and copy number variants (CNVs), the latter of which include losses of chromosome segments 1, 4, and 18 and gains in chromosome 8 [16,18]. Specifically, amplifications in 2q11.1, present in 100% of sensitive NS-TGCTs, were significantly associated with chemosensitivity [16]. In contrast, a recurrent small-scale focal gain at 2q32.1, encompassing gene *FSIP2*, was observed at 15–20% frequency in two independent TGCT cohorts of metastatic and resistant tumors [18,33,50]. Moreover, gains in 3p25.3 were associated with shorter progression-free survival and overall survival, with the strongest association observed in cisplatin-resistant NS, excluding pure teratoma [51] (Table 1). Large-scale CNVs (>1 Mb) show potential as biomarkers for chemotherapy response but need to be validated in larger study cohorts. Additionally, due to recent considerations about genomic heterogeneity between study populations, it is important to consider that most published works in TGCT have been done in European descendant populations; therefore, the concordance in findings could be limited by population differences that can be underlying intrinsic chemoresponse.

**Table 1 ijms-24-07873-t001:** Summary of genomic markers related to chemoresponse in TGCT.

	Marker	Type	Phenotype Associated	Sample Type	Assays	Main Results and Marker Function	Reference
Numeric variants	2q11.1	CNV (gains)	Sensitivity	Tumor tissue	WES	Amplifications were present in 100% of sensitive patients and not found in resistant tumors	[16]
	2q32.1	CNV (gains)	Resistance	Tumor tissue	WES	Amplification was correlated with refractory and metastatic tumors	[18]
	3p25.3	CNV (gains)	Resistance	Cell lines	Genomic profiling/qPCR	Gains were detected at low frequencies in primary tumors but at higher frequencies in inducted cisplatin-resistant tumors	[51]
Genes	*ERCC1*	↑exp	Resistance	Tumor tissue and cell lines	qPCR	Overexpression in both cell lines and tumor tissue is a finding in acquired resistant phenotypes	[52]
	*ERCC1*	↓exp	Sensitivity	Tumor tissue and cell lines	qPCR	Downregulation in cell lines and tumor tissue is a finding in sensitive phenotypes	[52]
	* HMGB4 *	depletion	Resistance	Cell lines	HMGB4 Knockout	Plays a major role in sensitizing TGCTs to cisplatin knockout cause differences in cell cycle progression following cisplatin treatment	[53]
	* HMGN5 *	↓exp	Resistance	Cell lines	Exp microarray	mRNA levels were remarkably upregulated in resistant subclones compared with the corresponding parental cells. Knockdown substantially reduced the viability of cisplatin-resistant TGCT cells in the presence of cisplatin	[54]
	*REV7*	↓exp	Sensitivity	Cell lines	qPCR	Depletion promoted chemosensitivity. In addition, inactivation in cisplatin-resistant TGCT cells meant they recovered chemosensitivity at almost equal levels to parental cells in vitro and in vivo	[55]
	*CCND1*	↑exp	Resistance	Tumor tissue	qPCR/IHC	Expression was significantly higher in resistant cases compared with sensitive samples	[56]
	*OCT4*	↓exp	Resistance	Tumor tissue and cell lines	qPCR/IHC	Decreased expression promotes higher differentiation, thus inducing a resistant phenotype	[39]
	* CTR1 *	depletion	Resistance	Cell lines	qPCR/WB	Increased protein expression was observed for the most cisplatin-sensitive cell lines, and depletion promotes a resistant phenotype	[57]
	*MDM2*	CNV (gains)	Resistance	Cell lines	qPCR	CNV gains induced a resistant phenotype through inhibition of the p53 pathway	[58]
	*MDM2*	↑exp	Resistance	Tumor tissue	IHC	Overexpression at tissue level in TGCT correlates with more aggressive phenotypes that tend to acquire resistance	[59]
	*KRAS*	CNV (gains)	Resistance	Tumor tissue	qPCR	Amplifications are associated with poor prognosis in 80% of cases	[60]
	A*KT1/PIK3CA*	somatic mutations	Resistance	Tumor tissue	WES	Somatic mutations are present with a high frequency exclusively in resistant tumors	[61]
	*TEX11*	↑exp	Resistance	Cell lines	Exp microarray	Gene silencing in cisplatin-resistant TGCT cells increased the percentage of double-strand break marker γH2AX-positive cells. Overexpression promotes resistant phenotypes	[54]
	*HIF-1* *α*	↓exp	Sensitivity	Tumor tissue	IHC	Low expression levels in TGCTs, specifically SE and mixed NS, promotes a sensitive phenotype	[62,63]
	*TDRG1*	↑exp	Resistance	Tumor tissue and cell lines	qPCR/IF	Overexpression regulates chemosensitivity to cisplatin in cell lines through PI3K/Akt/mTOR signaling and mitochondria-mediated apoptotic pathways both in vitro and in vivo	[64]
	*ALDH1*	↑exp	Resistance	Cell lines	qPCR	The ALDH inhibitor disulfiram restored sensitivity to cisplatin upon combinatorial treatment in both resistant cell lines and significantly inhibited tumor growth	[65]

Abbreviations: SE: seminoma, NS: no-seminoma, CVN: copy number variations, qPCR: quantitative PCR, WB: Western blot, WES: whole-exome sequencing, IHC: immunohistochemistry, IF: immunofluorescence, exp: expression.

### 4.1. Mechanisms of Cisplatin Sensitivity in TGCT

One of the principal features of TGCTs is their extraordinary responsiveness to treatments that induce DNA damage [17]. This intrinsic vulnerability can be explained as a consequence of two main variables; on one hand, the reduced capacity of the cell to repair DNA fragmentation induced by cisplatin, and, on the other, the loss of active pumps to capture cisplatin from the cell to prevent further damage [7].

#### 4.1.1. Cisplatin cytotoxicity

As a mechanism by itself, the activity of cisplatin is mediated by its effectiveness in binding to DNA to elicit intracellular responses. Cisplatin uptake is mediated by the copper membrane transporter (*CTR1*) [66]. Once inside the cell, the molecule hydrolyzes and becomes positively charged, allowing interaction with nucleophilic molecules inside the cell, including DNA and RNA. In the nucleus, cisplatin binds to purine residues, causing the formation of intrastrand and interstrand DNA crosslinks, leading to the inhibition of DNA transcription [41,67,68]. The altered structure of this molecule activates the DNA damage response (DDR) in the cell to repair damage before cell cycle progression, mainly through the nucleotide excision repair pathway (NER) [41]. However, some essential proteins involved in NER, like ERCC1 (Table 1), XPA, and XPF have been shown to have low expression levels in TGCT, resulting in an impaired capacity to repair DNA cross-links (Figure 1(AI)).

Interestingly, testis-specific high mobility group box (HMG) proteins, such as HMGB4, selectively bind to DNA–cisplatin adducts, rendering lesions inaccessible to the NER machinery, modulating sensitivity to the chemotherapeutic agent [13] (Figure 1(AI)).

*REV7* is a multifunctional gene with testis-specific expression that participates in multiple DNA repair pathways; their depletion causes chemosensitivity in acquired resistant models, which is associated with DNA double-strand break (DSB) accumulation leading to apoptosis activation. Inactivation of *REV7* in cisplatin-resistant TGCT cells recovered their chemosensitivity at almost equal levels to parental cells in vitro and in vivo [55] (Table 1). The combination of genomic instability and mutational defects explained by defective DSB repair allows more efficient induction of cell cycle arrest and apoptosis. In TGCTs, despite the fact that the G2/M checkpoint is functional, the G1/S checkpoint is deregulated, and therefore causes a premature entry into S phase, a temporary delay in phase S, and apoptosis after exposure to cisplatin [69], generating intrinsic hypersensitivity.

Cisplatin can also generate cytotoxicity by inducing oxidative stress, through the production of reactive oxygen species (ROS), responsible for DNA damage and mitochondrial dysfunction. This process leads to increased expression of the FAS death receptor (transcriptional target of p53) and the subsequent activation of the extrinsic apoptosis pathway [41,67] (Figure 1(AII)).

#### 4.1.2. Induction of Apoptotic Pathways

The hypersensitivity of TGCTs to cisplatin is driven by the induction of multiple apoptotic pathways, with wild-type (WT) p53 playing an important role [70]. Functional loss of p53 signaling leads to a lack of cell cycle regulation and is associated with more aggressive tumors and therapy resistance [71]. Paradoxically, in TGCTs, *TP53* mutations have rarely been observed [72] and often exhibit high expression of p53-WT in both the cytoplasm and the nucleus [73,74], suggesting that an intact pathway is more important for induction of the apoptotic response.

TGCTs with WT p53 have higher apoptotic potential with p53 transactivation of the *CDKN1A* gene, which encodes for the cyclin-dependent kinase (CDK) inhibitor p21 and is essential for inducing cell cycle arrest [75]. p21 expression has been shown to play a key role in cisplatin sensitivity; low levels of p21 were identified in SE and EC, whilst higher levels were found in teratomas [76] (Figure 1(AII)). However, more studies are needed concerning these apoptosis/cell cycle effectors [56].

It is well known that TGCTs largely retain the molecular characteristics of their embryonic cell precursors, such as expression of the pluripotency transcription factors *OCT4* and *NANOG* [77]. High levels of *OCT4* have been associated with higher expression levels of *NOXA,* which binds to anti-apoptotic Bcl-2 members and is involved in regulation of the intrinsic apoptosis pathway [78] (Figure 1(AI)). Cellular differentiation mediated by downregulation of *OCT4* induces cisplatin resistance in both TGCT and embryonic stem cells [79]. This phenomenon is consistent with clinical observations that less differentiated TGCT subtypes (with *OCT4* and *NANOG* overexpression) have a sensitive phenotype, whereas differentiated teratomas (with *OCT4* and *NANOG* subexpression) are more resistant to cisplatin [17]. This finding was also supported by Abada et al., who found that induced differentiation of EC germ line cells through a differentiating agent (retinoic acid) produced a reduction in pluripotency markers *NANOG* and *OCT3/4* and an acute increase in cisplatin and paclitaxel resistance within four days of treatment [39] (Table 1). 

### 4.2. Mechanisms of Cisplatin Resistance in TGCT

A minority of TGCTs exhibit cisplatin resistance [41], and treatment options for these cases are limited and long-term survival is poor [17]. According to Galluzi et al., the mechanisms of resistance to cisplatin can be classified as pre-target, on-target, and post-target [80]. Pre-target implies alterations that precede the binding of cisplatin to DNA and mechanisms that underlie the capture of cisplatin in the cell; on-target resistance implies alterations directly related to the response mechanisms to damage produced by DNA–cisplatin adducts, and, finally, post-target resistance implies mechanisms subsequent to DNA repair, which involve cisplatin-mediated induction of apoptosis (Figure 1(BII)). Following the pathway of cisplatin throughout the cell allows a better understanding of the mechanisms.

#### 4.2.1. Pre-Target: CTR1 Receptor Alterations Promotes Cisplatin Uptake Failure

Previous to exposure to the therapeutic agent, tumor cells can avoid the cytotoxic potential of cisplatin before it binds to DNA by at least two main mechanisms: (1) decreased intracellular accumulation of cisplatin and (2) increased cytoplasmic detoxification by cisplatin scavenger agents [80]. Even though these mechanisms could play an important role in cisplatin resistance in several malignant neoplasms, they have not been recognized as substantial determinants of cisplatin resistance in TGCT. The activity of cisplatin correlates with its ability to bind to DNA and elicit an intracellular response. Therefore, receptor alteration, persistence of the DNA repair system, or cellular tolerance can lead to chemotherapy resistance in these tumors. Within the first mechanism, the role of copper transporters seems the most important, both in uptake and efflux, with downregulation of *CTR1* (Table 1) and upregulation of *ATP7A/ATP7B* in cisplatin-resistant tumors, respectively [43] (Figure 1(BII)).

Pre-target mechanisms also control the upregulation of cytoplasmic scavenger factors, such as reduced glutathione, -GSH, which binds to cisplatin-DNA, inhibiting its action [81]. Despite evidence pointing towards mainly low levels of scavenger players in TGCTs, contributing to the chemo-sensitive phenotype, upregulation might be involved in cisplatin resistance [43]. Nevertheless, stronger clinical evidence is still lacking to date (Figure 1(BII)).

#### 4.2.2. On-Target: BRAF, ERCC1 and NER/BER Pathways

Effective DNA repair and increased tolerance of DNA damage are mechanisms thought to play an important role in cisplatin-resistant TGCT cells. Alteration of factors in the DDR pathways, as well as rare genetic mutations, result in greater efficiency in DNA repair (Figure 1(BII)). This has led to studies suggesting that TGCTs may be vulnerable to PARP inhibitors [14]. Interestingly, it has been proposed that the expression levels of some of these components are mediated by promoter methylation, providing a synergy between traditional DNA repair/DDR-based mechanisms of chemotherapy resistance and epigenetic reprogramming in TGCT [14,82]. A study reported differential abundance in 144 proteins between isogenic resistant and sensitive cell lines implicated in DDR pathways, confirming the regulation of key resistance-associated proteins (CBS, ANXA1, LDHA, CTH, FDXR) [83].

The primary repair mechanism utilized by cells following cisplatin-induced DNA damage is the NER pathway, with ERCC1 as an essential player to catalyze DNA excision and cisplatin resistance. The overexpression of *ERCC1* and *XPA* has been correlated with resistant TGCT [52] and other cisplatin-resistant neoplasms such as urothelial carcinomas [84] (Table 1).

Mutations in *XRCC2* have been reported in five refractory TGCT cases [17,85]. XRCC2 belongs to the RAD51 protein family and is involved in DNA repair via HR. Interestingly, *XRCC2* variants were found to support resistance to cisplatin-induced DNA damage and have been associated with breast cancer risk and survival [86].

Other important altered pathways are related to cisplatin resistance, such as proliferation and general stress response pathways, including the heat shock response [52,80] (Figure 1(BI)).

*BRAF* plays a role in regulating the MAP kinase/ERK signaling pathway, which affects cell division, differentiation, and secretion. *BRAF* mutations, which are correlated with microsatellite instability (MSI), have been associated with cisplatin resistance [87]. Honecker et al. compared control TGCTs to cisplatin-resistant TGCTs and found a significantly higher incidence of *BRAF* mutations in resistant vs. sensitive tumors (26% vs. 1%) [87]. However, a subsequent study in patients with cisplatin-resistant TGCT was unable to identify *BRAF* mutations, though hotspot variants in *PIK3CA*, *AKT1*, and *FGFR3* were noted [61].

In TGCTs, it is known that, despite the fact that the most differentiated tumors tend to present on-target resistance mechanisms, at the genomic level [18], somatic mutations studied in intrinsic chemoresponse do not seem to increase between different subtypes. Furthermore, cisplatin-resistant tumors do not possess a clear tendency of having more genomic abnormalities according to tumor progression [16].

#### 4.2.3. Post-Target: Pro-Apoptotic Pathway Dysfunction (P53, PI3K/AKT)

Variants in signaling pathway genes that mediate apoptosis in response to DNA damage represent an essential factor in TGCT resistance to cisplatin. Both underexpression and dysfunction of proapoptotic genes, as well as overexpression of antiapoptotic genes, can lead to aberrant induction of apoptosis. The role that p53 mutations play in TGCT chemoresistance is controversial, due to the low frequency of p53 mutations in a subset of resistant TGCTs [18,19,33]. Mutations in other components that regulate the p53 pathway could have more relevance in TGCT resistance to cisplatin than p53 itself.

Perhaps the most convincing mechanism for TGCT resistance due to acquired mutation is increased *MDM2* copy number [43] (Figure 1(BII)). MDM2 exerts its inhibitory role by nuclear export and ubiquitylation of p53, leading to its degradation [88,89]. *MDM2* amplification is likely a selective mechanism to prevent cell cycle arrest and DNA repair during the progression of disease, making this CNV an attractive therapeutic target, with multiple inhibitors that are currently undergoing clinical evaluation [90]. Nutlin-3 is an MDM2 inhibitor and causes an accumulation of p53 and up to a 10-fold increase in cisplatin sensitivity in NTERA2 [58]. *MYCN* amplifications, also affecting p53 signaling, have also been described in chemoresistant and refractory patients [88].

AKT overactivation was observed in cisplatin-resistant cells, and, subsequently, hyperactivation of the PI3K/AKT pathway increased phosphorylation of p21 (leading to its cytoplasmic accumulation) of *MDM2* (leading to inhibition of p53-mediated apoptosis). In addition, somatic mutations in the *AKT1* and *PIK3CA* genes have been reported exclusively in cisplatin-resistant TGCT [61] (Figure 1(BII)).

*FGFR3* and K*RAS* mutations were identified at a higher frequency in chemoresistant versus sensitive tumors [61]. K*RAS* mutations are more common in SE compared to NS [23] (Figure 1(BII)). Interestingly, *KRAS* is located on chromosome 12, which is frequently amplified in TGCTs [17]. *FGFR3* inhibition is already utilized clinically, and recent advances in targeted therapies have challenged the previously held notion that *KRAS* is an ‘undruggable’ target [91].

Finally, a cell cycle regulator has been described as another possible contributor to cisplatin resistance through the differential overexpression of *CCND1* (Cyclin D1) in cisplatin-resistant tumor samples. General *CCND1* expression was higher in cisplatin-resistant cases than sensitive ones, with no significant difference between SE and NS (Table 1).

The cisplatin-resistant phenotype can also be sustained by alterations in signaling pathways that are not directly engaged by cisplatin, so-called “off-target” effects. 

## 5. TGCT Epigenomic Hallmarks in Chemoresponse

The genomic features described in TGCT, especially the low mutational burden and the absence of alterations in driver genes, do not seem to fully explain the etiology of these tumors [14,33]. Taken together, these results have allowed the hypothesis that epigenetic regulation plays a main role in TGCT pathogenesis, even more so considering that these mechanisms respond dynamically to changes in the environment and the lifestyles of patients [14,92,93].

Epigenetic machinery can operate synergistically at different levels to modulate gene expression—DNA and RNA methylation, histone post-transcriptional modifications, and regulatory non-coding RNAs (ncRNAs)—despite adding greater complexity by involving more regulatory processes and corresponding effectors. The inclusion of these elements also expands the potential for discovering the underlying causes of unresolved phenomena, such as the one which prompted this study. The main contributions to describing the chemoresponse from this perspective are summarized as follows (Table 2) and integrated into Figure 1(AI,BI).

### 5.1. DNA Methylation

DNA methylation is mainly defined in mammals as the incorporation of a methyl group at carbon 5 of cytosine (5mC) in a CpG context, performed and maintained by DNA methyltransferase enzymes (DNMTs). DNA demethylation can occur in a passive way, through replication-dependent dilution in cellular division, or can be caused by TET enzymes by oxidizing 5mC. This epigenetic modification is involved in the maintenance of genomic stability, as well as in the regulation of gene expression [94].

DNA methylation is a key feature in deciphering the two faces of chemoresponse in TGCT. From a genomic perspective, high levels of DNA methylation are associated with cisplatin resistance, while global hypomethylation is associated with sensitivity [95,96]. It has been accepted that an active (open) chromatin state imposed by low methylation levels facilitates the integration of cisplatin into DNA [97] (Figure 1(AI)).

Hypermethylation of specific gene promoters, such as *RASSF1A, HIC1, CALCA, MLH1,* and *MGMT,* has been associated with cisplatin resistance [98,99] (Figure 1BI), whereas *RARB* promoter hypermethylation is associated with tumors sensitive to cisplatin (Figure 1(AI)) (Table 2). None of these genes has been validated as a DNA methylation-based biomarker for cisplatin response. Nevertheless, hypermethylation of *RASSF1A* in circulating cell-free DNA was described as an effective biomarker in TGCT diagnosis (sensitivity = 86.7%) [100].

Integrative transcriptome and methylome analyses of four independently derived isogenic cisplatin-resistant TGCT cell lines revealed a substantially higher number of hypermethylated CpG probes compared to hypomethylated CpG probes in cisplatin-resistant cells relative to parental, sensitive cells. Hypermethylation occurred mainly in repressive DNA segments, CTCF sites, and LADs, suggesting that resistance implies a whole nuclear reorganization of chromatin structure, even more so considering that CTCF binds to DNA in a methylation-dependent manner [101] (Figure 1(BI)). Gene loci that had suffered a bidirectional shift between gene promoter and gene body DNA methylation status were associated with downregulation of tumor suppressor genes and upregulation of polycomb targets [96]. Another approach, employing the 850K EPIC methylation array in samples from four TGCT patients with distinct responses to cisplatin, found that differentially hypermethylated promoters in cisplatin resistance were enriched in pathways related to regulation of the immune microenvironment (chemoattraction), as were differentially hypomethylated promoters in pathways related to DNA/chromatin binding and regulation, supporting the possibility of investigating chromatin remodelers as chemoresponse mediators [102].

DNA methylation levels are associated with state of differentiation and the response to cisplatin. Seminomas show an undifferentiated phenotype, global DNA hypomethylated status, and excellent response to therapy. In contrast, non-seminomas are characterized as showing increasing DNA methylation levels, differentiated status, and a decrease in cisplatin sensitivity [19,103]. These suggest that open-pluripotent chromatin may possess inherent transcriptional plasticity that grants a faster response to DNA damage in TGTC, compared to other somatic malignancies [13]. SEs display lower expression levels of DNMTs and higher expression levels of TET2 compared to NSs [104,105,106] (Figure 1), which explains their global hypomethylated pattern and the success of DNMT inhibitors, which seem to rescue sensitivity to chemotherapy with cisplatin in TGCT cell lines [107,108]. Besides, in accordance with TET enzyme catalytic activity, SEs show low levels of 5hmC, whereas high levels are found in differentiated teratomas, ECs, and yolk sac tumors [109].

In summary, further efforts are required to account for the clear differences in methylation profile between therapy-resistant and sensitive samples; however, these data support the role of DNA methylation as a key factor in the chemoresistance of TGCT and the implementation of targeted hypomethylating treatments for resistant patients.

### 5.2. RNA Methylation

N6-methyladenosine (m^6^A) is the most abundant modification among mRNAs and has been shown to be enriched near stop codons and 3′ untranslated regions (3′UTR) [110]. This dynamic mark can be established by ‘writer’ enzymes (e.g., METTL3, METTL14), removed by ‘eraser’ enzymes (e.g., FTO and ALKBH5), and interpreted by ‘reader’ enzymes (e.g., YTHDC1–2, YTHDF1–3, IGF2BP1–3). m^6^A has been linked to the control of mRNA stability, modulation of alternative splicing, translational efficiency, RNA structure for protein binding, and pri–miRNA processing [111]. Deregulation of this machinery has important implications for tumorigenesis and the progression of several neoplasms [112,113].

Epitranscriptomics, which involves chemical modifications in RNA, is a promising and emerging research area looking for novel therapeutic targets [114]. Unfortunately, information related to TGCTs and their response to therapy is reduced to a few publications [115,116].

**Table 2 ijms-24-07873-t002:** Summary of epigenetic features associated with the different phenotypes of cisplatin response.

Marker	Role in Response	Mechanism of Resistance	Sample Type	Assays	Main Results and Marker Function	Reference
*RASSF1A* (↑5mC)	Resistance	Intrinsic	NS tissue (31Se; 39Re)	qMSP	(52% Re vs 28% Se) Negative regulator of cell growth	[98]
*HIC1* (↑5mC)	Resistance	Intrinsic			(47% Re vs 24% Se) Transcription factor that acts as a tumor suppressor	
*RARB* (↑5mC)	Sensitivity	Intrinsic			(0% Re vs 14% Se) Receptor involved in morphogenesis, cell growth and differentiation	
*MGMT* (↑5mC)	Sensitivity	Intrinsic			(31% Re vs 13% Se) MGMT is a DNA repair enzyme	
*CALCA* (↑5mC)	Resistance	Intrinsic	TGCT tissues (47Se; 15Re)	qMSP	47.4% (9/19, *p* = 0.005) of samples with methylated loci presented refractory disease, also associated with NS tumors. Gene is involved in calcium regulation, acts as a vasodilator	[99]
*MGMT* (↑5mC)	Resistance	Intrinsic			38.1% (08/21, *p* = 0.067) of tumors presenting MGMT methylation were refractory, which was also associated with NS histology	
Global ↑5mC	Resistance	Acquired	NT2/D1, 833K, and 2102EP and cisplatin-resistant sublines	EPIC 850 K array and RNA-Seq	Acquired cisplatin resistance in TGCT triggers net ↑5mC. Hypermethylation in resistant cells is associated with repression of cancer suppressor genes and nuclear organization of repressive chromatin, while hypomethylation is associated with the polycomb pathway	[96]
Global ↑5mC	Resistance	Acquired	Matched primary and metastatic tissue from four patients	EPIC 850 K array	Hypermethylation in promoters of genes related to regulation of the immune microenvironment. Hypomethylation of promoters on pathways related to DNA/chromatin binding	[102]
*VIRMA* (↑exp)	Resistance	Acquired	TCam-2, NCCIT, 2102Ep, and NT2 and cisplatin-resistant sublines	RT-qPCR, ELISA and dot blot (m^6^A quantification). CRISPR/Cas9 (knockdown of VIRMA) followed by cell viability, proliferation, invasion, and CAM assays.	The component of the m^6^A writer complex VIRMA contributes to tumor aggressiveness and to cisplatin resistance, both in vitro and in vivo*,* by regulating DNA damage response	[115]
miR-371-373 cluster	Resistance	Acquired	NTERA-2, NCCIT, and 2102EP, and cisplatin-resistant sublines	RT-qPCR and LDA	Upregulated in NTERA-2 and NCCIT resistant cells; possibly promotes resistance by counteracting wild-type p53-induced senescence	[117]
hsa-miR-99a/-100/-145	Resistance	Acquired			About 10-fold down-regulated in NTERA-2- and NCCIT resistant clones	
miR-302a	Sensitivity	Acquired	NTERA-2 and its cisplatin-resistant subline	Overexpression via transfecting vector, RT-qPCR (expression). Cell proliferation and drug-sensitivity assay	Up-regulation of miR-302a significantly increased the sensitivity of NT2 cells to cisplatin by enhancing cisplatin-induced G2/M phase arrest and the subsequent progression to apoptosis	[118]
miR-302 cluster	Resistance	Acquired	NT2-D1, 833 K, and cisplatin-resistant sublines	Inhibitor-mediated transient transfection. RT-qPCr (expression). Cell survival, proliferation, and invasion assays	miR-302s act as TGCT oncogenes by inducing the expression of *SPRY4* and activating the MAPK/ERK pathway while inhibiting apoptosis	[119]
miR-383	Sensitivity	Acquired	NTERA-2 and its cisplatin-resistant subline	miR-383 mimics and miR-383 inhibitor transfection, RT-PCR/WB, cisplatin sensitivity assay	This miRNA ↓*PNUTS* levels; this blocks the phosphorylation of H2A and induces cell cycle arrest	[120]
Molecular signature: miR-218-5p, miR-31-5p, miR-375-5p, miR-517-3p, miR-20b-5p and miR-378a-3p	Resistance	Acquired	Discovery: Cisplatin-sensitive and -resistant TGCT cell linesValidation: TGCT tissue (*n* = 53) and control (*n* = 33)	miRNA microarray profiling (discovery), RT-qPCR (validation)	New panel of biomarkers for better prediction of chemoresistance and more aggressive phenotypes	[121]
↓H3K27me3 (polycomb activity)	Resistance	Acquired	NT2/D1, 833K, and 2102EP and cisplatin-resistant sublines	RNA-Seq and GSEA. Drug Tx with GSK126 (EZH2 inhibitor) and GSKJ4 (JMJD3 inhibitor). Cell viability and proliferation assays	Resistant lines express genes normally repressed by polycomb. Repression of H3K27me3 conferred cisplatin resistance to parental cells while induction of the mark resulted in increased cisplatin sensitivity in resistant cells	[122]
Crosstalk: ↓DNMT3B → ↑H3K27me3	Sensitivity	Acquired	5-aza-resistant cell lines	Drug Tx 85-aza) and cell viability and proliferation assays. Lentiviral shRNA (DNMT3B knockdown) followed by RNA-Seq	DNMT3B knockdown alone in parental cells resulted in increased expression of H3K27me3, *EZH2*, and *BMI1*, and conferred 5-aza resistance and cisplatin sensitization. Patients resistant to cisplatin may have high levels of *DNMT3B* and *KDM6B* and low levels of H3K27me3	[108]
↑H2Bub1 in Lys120	Resistance	Acquired	NCCIT and 2102EP and cisplatin-resistant sublines	WB for H2Bub1 levels with and w/o ATRA Tx. H2Bub1 knockdown followed by MTT and colony formation assay	↑H2Bub1 levels in resistant cells; inhibition of H2Bub1 formation impaired DNA repair and decreased cellular survival (enhanced sensitivity)	[123]

Abbreviations: SE: seminoma, NS: no-seminoma, Se: sensitivity, Re: resistant, qMSP: quantitative methylation-specific polymerase chain reaction, LDA: low-density array, RT-qPCR: real-time quantitative PCR, WB: Western blot, GSEA: Gene set enrichment analysis, Tx: treatment, CAM: chorioallantoic membrane, 5mC: methylation.

METTL3, the first m^6^A writer identified, is the only catalytic subunit of the methyltransferase complex (MTC), formed by other indispensable components for substrate recognition and stability, such as METTL14, WTAP, and VIRMA [124]. In TGCTs, METTL3 promotes resistance of the seminoma-like TCam-2 cell line by inducing m^6^A methylation in TFAP2C transcripts; this modification is recognized by the reader IGF2BP1, promoting RNA stability. Apparently, m^6^A methylated-TFAP2C activates DNA repair genes *WEE1* and *BRCA1*, affecting the cellular response to cisplatin treatment response in SE [115,116] (Figure 1(BII)). Overexpression of *METTL3* promotes proliferation, migration, and invasion in NCCIT and TCam-2 cells [125], reinforcing the oncogenic role of this writer in TGCT.

The abundance of m^6^A and expression of *VIRMA*/*YTHDF3* are different among TGCT subtypes, with higher levels in SE, suggesting that this writer/reader pair cooperates to induce m^6^A modification and maintains the SE phenotype [126]. In mESC, m^6^A methylation is inversely correlated with mRNA stability; it has been proposed that destabilization and further degradation of transcripts encoding developmental regulators allows the maintenance of the stemness.

Miranda-Gonçalves et al. described the role of VIRMA in the acquisition of cisplatin resistance. Their CRISPR/Cas9-mediated knockdown of VIRMA in NCCIT cells showed reduced abundance of m^6^A levels and significant diminution of viability after cisplatin exposure, an observation confirmed in vivo with *VIRMA* knockdown tumors. The enhanced response to cisplatin in NCCIT cells after *VIRMA* knockdown was related to a significant increase in DNA damage (with higher expression of γH2AX and *GADD45B*) and downregulation of *XLF* and *MRE11* genes involved in DNA repair [115] (Table 2).

Specific, differentially expressed m^6^A RNA methylation regulators between tumors and normal tissues have made it possible to generate risk signatures that can predict progression-free survival rates [127,128]. It is hoped that by refining understanding of the role of modifiers in TGCT, these signatures can be used to predict response phenotypes to different drugs.

### 5.3. Non-Coding RNAs

microRNAs (miRNAs) and long noncoding RNAs (lncRNAs) are two major classes of non-coding transcripts. miRNAs are short RNA molecules that negatively regulate transcript stability and translation [129]. lncRNAs are non-coding genomic transcripts longer than 200 nucleotides that can modulate chromatin structure and affect RNA splicing, stability, and translation, among other mechanisms [130]. Both molecules regulate fundamental cellular processes, and their abnormal expression is critical to the pathogenesis of human disease; they represent promising non-invasive serum biomarkers and therapeutic targets. The general roles of miRNAs and lncRNAs in testicular cancer have started to be understood [131,132].

In TGCTs, two main clusters of miRNAs are overexpressed: the miR-371–373 and miR-302 clusters [93]. Port et al. reported increased expression of the miR-371-373 cluster in cisplatin-resistant TGCT sublines and surmised that its role in resistance is to counteract p53-induced senescence [118]. Currently, it is well known that this cluster regulates activation of the Wnt/β-catenin pathway and that miR-371a-3p, one of the components of the cluster, has been positioned as a powerful marker for diagnosis, staging, and prognosis in TGCT. However, regarding its participation in response to therapy, studies have focused only on positioning it as a biomarker for the early diagnosis of relapse [133,134]. On the other hand, results around the mir-302 cluster have been controversial, since, in 2013, Liu et al. proposed that upregulation of mir-302a significantly increases the sensitivity of NT2 to cisplatin [119], while Das and collaborators suggest that miR-302s act as TGCT oncogenes by inducing the expression of SPRY4 and inhibiting apoptosis via increased surviving expression [120] (Table 2).

An interesting contribution on this subject occurred when the cell viability of T-Cam sensitive cells cultured with exosomes released by TCam-2 cisplatin-resistant subclones was tested. miR-193b-3p, enriched in the exosomes, confers enhanced cisplatin tolerance to TCam-2 cells by targeting the transcription factor ZBTB7A, which further decreases apoptosis and promotes cell cycle progression [135].

Recently, Roška et al. proposed an entire panel of biomarkers for the prediction of chemoresistance and aggressive phenotypes in NS. The panel consists of miR-218-5p, miR-31-5p, miR-375-5p, miR-517a-3p, miR-20b-5p, and miR-378a-3p. Although miRNAs can bind to multiple targets, four predicted targets shared among all miRNAs were found: *ZBTB20, FZD4, CACUL1,* and *CEP85L*, establishing the possibility that the functions of these mRNAs could contribute to the development of cisplatin resistance in TGCTs [122] (Table 2).

Regarding cisplatin sensitivity markers in TGCT, it has been established that overexpression of miR-383 sensitizes NT-2 cells to cisplatin mir-383 targets PNUTS and IRF-1 in embryonic carcinomas. Depletion of PNUTS impairs the phosphorylation of γH2AX, the histone variant involved in DNA repair, causing cell cycle arrest and induction of apoptosis [121] (Figure 1(AI)).

*H19*, a long non-coding RNA upregulated in the cisplatin-resistant TCam-2 cell line promotes cisplatin resistance by sequestering miRNA-106b-5p. This miRNA decreases the expression of TDRG1, a recognized oncogene exclusively expressed in testis [121]. This mechanism facilitates cell survival in cisplatin-based chemotherapeutic conditions [136]. Interestingly, *H19*, along with four other lncRNAs (*NEAT1, PVT1, SFTA1P, TRPM2-AS*), was overexpressed in subtype 2 seminomas [10]. These five lncRNAs are already identified as responsible for cisplatin resistance in gastric, lung, and ovarian cancer [137].

Additional validation studies in larger cohorts of patient samples are required to validate the involvement of these lncRNAs as biomarkers for cisplatin response. In addition, there are other miRNAs with increased expression in TGCT that could justify investigation of their role in chemoresponse: in the TCam-2 and 2102Ep cell lines, miR-223-3p negatively regulates the expression of the tumor suppressor FBXW7 [138]; this miRNA has already been associated with cisplatin resistance in human gastric cancer cells [139]. Furthermore, miR-885-5p (a p53 activator) can be highly elevated in mature teratomas, which are practically resistant to cisplatin [140].

### 5.4. Histone Post-Transcriptional Modifications

N-terminal histone tails can acquire post-translational modifications that have different effects depending on which residue is modified and the chemical group added. This intricate “code” acts in a combinatorial and orchestrated manner to alter the structure of chromatin, ultimately impacting the accessibility of DNA to different enzymatic complexes that control gene expression [141]. Various human pathologies develop due to an altered pattern of histone post-transcriptional modifications [142,143].

Singh et al. described a decrease in global H3K27me3 as a mechanism of acquired cisplatin resistance in TGCT. Transcriptional profiling of cisplatin-resistant cells showed highly significant upregulation of genes normally repressed by polycomb repressive complex 2 (PRC2), associated with a decrease in the expression of BMI1 and EZH2. Moreover, inhibition of EZH2 conferred cisplatin resistance to parental cells, while induction of H3K27 methylation with the histone lysine demethylase inhibitor GSK-J4 resulted in increased cisplatin sensitivity in resistant cells [123]. It was recently shown that the increased expression of PRC2 target genes is also due to a decrease in the methylation of their promoters [96], establishing complex crosstalk between DNA methylation and H3K27me3 (Table 2). DNMT3B may be a critical upstream driver of this epigenetic crosstalk, because DNMT3B knockdown results in the induction of H3K27m3, EZH2, and BMI1 expression and confers 5-aza resistance in cell lines. Singh and collaborators predict that patients resistant to cisplatin may have high levels of DNMT3B and KDM6B, the H3K27me3 demethylase. Hypomethylation therapy with or without cisplatin and KDM6a inhibitors are promising strategies to hypersensitize TGCTs [108]. Furthermore, contrary to what has been observed in the treatment of ovarian, lung, and breast cancer, the use of EZH2 inhibitors may not have beneficial effects in the treatment of these patients [144].

In an NS cell line model, monoubiquitination of Lys120 of histone H2B (H2Bub1) has been linked to acquired cisplatin resistance. This mark facilitates the DNA damage response (DDR) and is involved in tissue differentiation, such that the deletion of H2Bub1 improves sensitivity to genotoxic treatment [124]. Interestingly, levels of the E3 ligase RNF20/40 complex are lower in SE compared to normal testes [145] and NS tissues [106], which could partially explain its strong response to treatment (Figure 1(BI))

Little is known about this level of epigenetic regulation, highlighting the urgent need for studies describing the actual genome-wide distribution of histone marks in TGCT. Following the proposal in this work, in which it is necessary to address sensitivity as a key to overcoming resistance, it will be important to further research the mechanisms by which seminomas maintain their undifferentiated state. This includes, for example, the abundance of H4/H2A R3me2 compared to NS [146], which is believed to repress genes involved in somatic differentiation programs, and the effects of the decrease in bivalent marks in the most differentiated tissues, as proposed by Singh et al. in their “Rock and a hard place” model [14]. Bivalent marks, usually found in development regulators, allow a more dynamic response to cisplatin (Figure 1(AI)).

In addition, SEs display higher expression levels of enzymes that establish activating modifications, such as KDM4D, KDM3A, KMT2B/C/D, SETD1A, and most classes of histone acetyltransferases (HATs) (Figure 1(AI)) compared to NSs, which overexpress HDACs [106] (Figure 1(BI)). Focused research on epigenetic modifiers with more differential expression patterns among TGCT subtypes is another opportunity to understand cisplatin resistance.

### 5.5. Integrative Landscape

In an epigenetic-focused description of the molecular landscape of TGCTs, it is hypothesized that traditional mechanisms of chemotherapy resistance, such as dysregulation of DNA repair and apoptotic response, could in fact be consequences of the altered epigenetic states of tumors through regulation of the expression levels of their main players. Nevertheless, it cannot be denied that genomic alterations can influence the activity of the epigenetic machinery. Grasso et al. described robust associations between variants in *MTHFR* (an essential enzyme for the synthesis of the methyl donor S-adenosylmethionine) and TGCT [147] (Figure 1(BI)); additionally, Jhuang et al. found *SINCAF*, a subunit of the Sin3/histone deacetylase complex (HDAC) hidden on chromosome 12p. This discovery becomes relevant considering that i(12p) is the most distinctive genomic mark among TGCTs, and the high levels of SINCAF that cause this amplification may favor the use of HDAC inhibitors (HDACi) as an alternative therapy in this neoplasia [148].

Although this chapter has described the epigenetic markers associated with chemoresponse in TGCTs, it is crucial to acknowledge that there is still much uncharted territory in this epigenetic landscape, but this represents an opportunity, especially in terms of utilizing epi-markers to identify promising therapeutic targets for precision medicine and fighting against cisplatin resistance.

## 6. Potential Approaches of TGCT Therapy

Treatment of refractory TGCTs is a major challenge due to their large inter- and intra-tumoral histological and biological variations. Unfortunately, research on them has been discouraged due to their favorable prognosis. Various chemotherapeutics with distinct cytotoxicity mechanisms have been explored to treat refractory TGCTs, typically in small, single-arm, phase II trials [41,43]. Single-agent and combination chemotherapy approaches have both been utilized, but few patients have obtained complete or lasting remission. Combination chemotherapy may be more successful than single-agent treatments. However, cure is rarely achieved for patients with cisplatin-refractory disease, and, for those who respond well and become resectable, long-term survival is achieved only in 10–15% of cases [41,43,149]. Oing et al. examined the efficacy of cabazitaxel in TGCTs. After a median follow-up of 23 weeks and administration of a median of two cycles, results showed a 12-week progression-free survival (PFS) rate of 31%. A total of 15% of patients had objective responses, and 23% experienced a tumor marker decline greater than 50%. The overall disease control rate was 39%. Cabazitaxel was well tolerated, however; dose reduction was rarely needed, with only 15% of patients requiring it. The previous data suggest that cabazitaxel has limited activity in heavily pre-treated TGCT patients. Currently, a prospective evaluation of cabazitaxel in multiple relapsed TGCTs is being conducted in two phase II trials (NCT02115165, NCT02478502). Analogously, a phase III trial is being set up comparing conventional paclitaxel, ifosfamide, and cisplatin (TIP) chemotherapy to high-dose carboplatin and etoposide therapy as a first salvage treatment for relapsed/refractory patients (NCT02375204), but outcomes are not yet available. Reliable biomarkers could help researchers understand cisplatin resistance, identify cisplatin-refractory disease, and successfully combine several pathway-targeting therapeutic approaches. Novel delivery options like epidrugs and immunotherapy could reduce associated toxicities and improve testicular cancer management [41,43,149,150].

### 6.1. Epridrugs

As discussed above, diverse studies have highlighted the role of epigenetics in TGCTs, with its influence on cisplatin resistance. Single-agent epidrugs, though promising in vitro and pre-clinical studies, have not been successful in clinical trials yet. Combining different treatment pathways may yield more satisfactory results for patients with poor prognoses, but safe delivery systems need to be developed to reduce toxicities [43,149].

Hypomethylating agents (HMAs) and HDACi have demonstrated potential results in in vitro and pre-clinical in vivo studies with monotherapy or combination therapies. HMAs (decitabine, guadecitabine, and 5-azacitidine) induced expression of tumor suppressor genes and p53 activation, encouraging a proapoptotic response and resensitizing TGCT cells to cisplatin. Low concentrations of guadecitabine inhibited progression and regressed cisplatin-resistant testicular cancer cells; this effect appears to be the result of p53 target induction, immune-related pathway induction, and pluripotency gene repression. A Phase I study using guadecitabine (30 mg/m^2^ × 5 days and cisplatin 100 mg/m^2^ in a 28-day treatment cycle) to sensitize relapsed TGCT patients reported three responses in 14 patients, with two complete responses lasting 5–13 months and 2–26 months for OS. ORR was 23%, with three stable diseases. The clinical benefit rate was 46%. Besides being well tolerated, neutropenia, thrombocytopenia, and anemia were common adverse events [41,43,151]. Similarly, Oing et al. examined the effects of DNA HMA 5-azacitidine (5-aza) on two embryonal cancer cell lines as well as on their cisplatin-resistant variants. 5-aza inhibited cell viability and progression by decreasing cellular survival and inducing apoptosis at low nanomolar doses in both cisplatin-sensitive and resistant cells. In the resistant cellular lines, combining 5-aza with cisplatin produced even better results, implying that combining DNMTs with chemotherapy may be a viable strategy for treating patients with refractory TGCTs [152]. In parallel, Lobo et al. evaluated the synthetic flavonoid drug MLo1302, a DNMT inhibitor, discovering that it decreased cell viability even in cisplatin-resistant cell lines, with an inhibitory concentration of 50 (IC50) falling within the nanomolar range for NCCIT and NTERA-2 cells. This shows that MLo1302 partially affected cell differentiation by lowering the protein expressions of pluripotency markers; this is comparable to decitabine, which causes DNA damage, promotes p53 activation, downregulates pluripotency factors, and restores sensitivity to cisplatin. However, further study of cisplatin resistance should be conducted to demonstrate that this treatment can also be clinically beneficial [43,153].

On the other hand, HDACis, such as romidepsin and trichostatin A, have exhibited anti-cancer activity in vitro and in vivo for TGCTs by inducing apoptosis, reducing tumor size, and preventing proliferation and angiogenesis [41]. On the same lines, animacroxam is especially effective in reducing both tumor growth and angiogenesis. It was found to be as effective as cisplatin, making it a viable alternative if patients cannot be treated with traditional cisplatin-based chemotherapy or have developed resistance to cisplatin [154]. Consistent with this, in research by Lobo et al., belinostat and panobinostat were examined for their impact on TGCT cell lines that were sensitive and resistant to cisplatin; 261 patient samples showed variable HDAC expression across cell lines, with low nanomolar IC50 values. Treatment with both medications reduced acetylation, caused cell cycle arrest, decreased proliferation, lowered Ki67 index, and elevated p21, while enhancing apoptosis. However, more research should be done to determine their potential as solo or combination agents [155].

BET inhibitors and ARID1A inhibitors have also shown potential antitumor effects in vitro and could potentially be used in combination with demethylating agents. For example, by studying the molecular function of ARID1A, Kurz, et al. were able to discover that this protein is involved in regulating transcription, DNA repair, and epigenetic stability. Their findings suggest that ARID1A may be a promising target for synthetic lethality and combination therapy to treat mutations in GCDS genes. By disabling ARID1A with CRISPR/Cas9, they were able to increase the sensitivity of TGCT cells to treatment with romidepsin and other antitumor agents [156]. Similarly, small-molecule inhibitor JQ1 has been found to be effective in tumor therapy, leading to DNA damage and cell cycle arrest in TGCT cellular lines. EC cellular lines responded to it, displaying a decrease in the factors of pluripotency and induction of mesodermal differentiation. TCam-2 cells had a higher tolerance for JQ1 and showed no evidence of differentiation. Treatment with JQ1 reduced tumor size, proliferation rate, and angiogenesis in ECs xenografted in vivo, and its combination with romidepsin allowed lower doses and less frequent application compared to monotherapy, indicating this therapeutic scheme as a potential novel therapeutic option for mixed TGCTs [155,157].

The combination of HDACi and BETi as dual inhibitors (LAK-FFK11, LAK129, LAK-HGK7) also decreased cell viability, caused apoptosis, and changed the cell cycle in cisplatin-resistant TGCT and other urological malignancies according to Burmeister et al., lowering cell viability both in vitro and in vivo, with TGCTs exhibiting the greatest reduction, while being less effective in non-malignant cells [158].

Alternatively, seven epi-drugs were chosen by Muller et al. based on their cytotoxicity at nanomolar (Quisinostat, JIB-04, Chaetocin, and MZ-1) to micromolar (LP99, PRT4165, GSK343) concentrations, demonstrating that epigenetic inhibitors may be beneficial for treating cisplatin-resistant sublines. Most of them produced either apoptosis or cell cycle arrest in TGCT cell lines, while fibroblasts had minimal response. Initial screening of other urological malignancies also yielded promising results [92].

Simultaneously, four potential drugs (PCNA-I1, ML323, T2AA, and MG-132) were tested to try to overcome cisplatin resistance. They were tested to evaluate differential mRNA expression profiles in in vitro cisplatin-resistant TGCT cell lines using NanoString technology. Van Helvoort et al. uncovered many genes involved in DNA repair and cell cycle control that were differentially expressed in the NTERA-2R cell line, discovering that MG-132 was cytotoxic even at nanomolar concentrations for all cell lines tested, and that it increased sensitivity to cisplatin. As a result, MG-132 might be a promising novel treatment for TGCT resistance [159].

In summary, epigenetic therapy might be promising for the treatment of TGCT in refractory disease or resistance to cisplatin-based therapy.

#### Therapy Enhancement by Epidrugs

A combination of epigenetic therapy and immunotherapy may also be a viable treatment option, as epigenetics is known to regulate the immune system, even though immunotherapy has only shown preliminary results, promising partial responses in some cases. Recently, CAR T-cell therapy was studied as a unique approach for neoplasms that do not react to immune checkpoint inhibitors and for those that are less immunogenic, with promising results in 13 TGCT patients. However, more research and a deeper understanding of the immune system’s activation processes is needed to validate its function as an immune checkpoint inhibitor therapy for TGCTs [41,43,160]. Epidrugs, combined with immunotherapies and cisplatin, could enhance tumor death, even for cisplatin resistant TGCTs. Further genetic, histological, and immunohistochemical research is required to demonstrate predictive variables controlling immunotherapy response. New trial designs should combine these novel drugs with the conventional standard of care in order to target benefits in early lines of treatment and assess their value in tackling this neoplasia [43,160,161,162].

Alternately, radiotherapy is a treatment option for particular cases of testicular cancer, which can also be used as a palliative modality or in recurrent disease. Even though it has not yet been examined in testicular cancer, radio-priming is a potential method of stably changing the gene expression pattern of a tumor to make it more susceptible to subsequent treatment. To do this, treatments that epigenetically modify cells, such as epigenetic inhibitors, can be combined to reset the cancer’s memory. Data from clinical studies in other types of cancer have shown that episensitization and the simultaneous administration of multiple anti-cancer treatments is an important anti-resistance strategy. Thus, radiotherapy should be considered as part of a potential regimen [163,164].

### 6.2. Other Targeted Therapies (OTTs)

Novel targeted medications, despite not falling within the category of epigenetic treatments, are considered potential substitutes for or combination agents with cisplatin. Although the use of OTTs in treating TGCTs has not yet established itself, due to low patient relapse rates and population diversity, some of them have shown promising in vitro action and have been studied in small, non-randomized phase I/II trials. Additionally, research in a patient population with cisplatin-resistant TGCTs suggests potential results with OTTs [41,43]. The following section lists some of the most relevant.

RTK inhibitors have a promising impact in TGCTs, due to their overproduction of vascular endothelial growth factor (VEGF), platelet-derived growth factor (PDGF), and their related receptors, which activates the AKT signaling pathway. Sunitinib and pazopanib can reverse TGCTs’ resistance to cisplatin. Castillo-Avila et al.’s study revealed tumor growth inhibition in TGCTs treated with sunitinib. A phase II clinical trial of sunitinib as a single agent showed a median PFS of 2 months and an OS of 3.8 months, with a total response rate of 13% partial remissions. In another phase II trial on five heavily pre-treated patients, one was free of disease progression to sunitinib for 12 weeks [41,165]. Similarly, Juliachs et al.’s study evaluated pazopanib’s efficacy with and without lapatinib in mouse orthotopic models, which all failed to respond to cisplatin but showed positive results with pazopanib’s anti-angiogenesis properties [41,166]. On the other hand, the use of palbociclib has been demonstrated in both in vitro and in vivo experiments to decrease the viability of TGCT cells when it is given in combination with cisplatin. With exposure to both these drugs, it was discovered that there was an additive impact compared to either medication alone in terms of postponing cell recovery from the toxic insult. In addition, it was found that cisplatin caused cell accumulation primarily in G2/M compared to the effect of palbociclib. Due to this, further analysis should be conducted to prove efficacy in a cisplatin resistance context [167].

Other potential therapeutic drugs (LGK-974 and PRI-724) were investigated by Schmidtova et al. in cisplatin-resistant TGCT cell lines as Wnt/β-catenin signaling inhibitors. They found that TCam-2 CisR and NCCIT CisR cell lines had higher levels of β-catenin and cyclin D1 but that NTERA-2 CISR cell lines had lower levels. While the pro-apoptotic effects of PRi-724 were seen in all cell lines, LGK974 had either a negligible or no effect. Their results suggest that CisR cells that are cisplantatin-resistant have altered Wnt/-Catenin signaling, and more research into pathway blocking in TGCTs is required [168]. Finally, metformin has been reported to inhibit cells in the G1 phase by activating phosphorylated *YAP1* and decreasing levels of cyclin D1, *CDK6*, *CDK4,* and RB, enhancing cisplatin chemosensitivity and inducing apoptosis. G1 phase arrest was observed in TCam-2 and NTERA-2 when metformin was withdrawn for 24 h. Combination treatment with metformin and cisplatin enhanced the anticancer effects, showing a potent reduction of cell proliferation as evidenced by decreased nuclear abundance of cyclin D1. Metformin also decreased the expression of *IGFBP1*, *IGF1R,* and *MMP-11*, which led to a decrease in HMGA1 abundance [169,170].

In conclusion, clinical research beyond phase I/II has been delayed due to non-fructiferous outcomes that directly impact on therapy response. However, in addition to the different precision medicine approaches, epidrugs appear to be the goal to strive for to effectively treat chemoresistant patients. The next step that must be given priority is to concentrate on identifying potential targets to develop OTTs and thus restore sensitivity to chemotherapy and radiation by epidrugs, in light of the current results (Table 3).

## 7. Driving the Future of TGCT Therapy

Cisplatin resistance is a multifactorial phenomenon that remains a central problem in TGCT clinical management. This review has focused on describing the genomic and epigenetic profiles of sensitive and resistant tumors; these characteristics might be seen as an opportunity to restore a “sensitive-like” profile in refractory patients and improve their response to therapy through modification of the epigenetic landscape. While DNMTs and HDACs inhibitors are the most extensively studied treatments in TGCT and other cancer types [41,164], the molecular profiles of sensitive and resistant tumors (Figure 1) demonstrate that these enzymes are not the only targets that need further research.

To detail the proposed strategy, we suggest a model that initially divides all clinical cases of TGCT into two groups based on their response phenotype to cisplatin-based therapy (Figure 2). The major group comprises tumors that are sensitive to therapy, where the epigenetic profile of open chromatin facilitates DNA breakage induced by cisplatin administration, resulting in cell division blockade and imminent apoptotic cell death. Unfortunately, in some cases, exposure to cisplatin triggers changes in the epigenome, leading to a closed chromatin state, transcriptional repression, and a loss of pluripotent features, especially in seminomatous or EC histologies. The whole process culminates in the acquisition of a resistant phenotype, in which case the cells are prepared to evade the cytotoxic effects of cisplatin. With the appearance of refractory tumors, we speculate that the optimal clinical strategy would be to administer cisplatin in combination with epidrugs, which could slow down the progressive and accelerated progression of cells toward a resistant state. This approach would enhance the effects of cisplatin, allowing the use of lower doses of the genotoxic drug and reducing its long-term side effects.

On the other hand, the second minority group consists of TGCTs that exhibit intrinsic resistance to cisplatin, where a restrictive chromatin conformation has already been established and limits the action of cisplatin. If we can find intrinsic resistance biomarkers that can identify these patients before administering chemotherapy, this would prevent them from receiving cisplatin treatment that would have no effect on tumor progression and would only result in negative long-term systemic effects. In these patients, epidrugs can provide a possible solution by re-sensitizing tumor cells, allowing them to acquire an epigenetic profile similar to intrinsically sensitive tumors. Once this reprogramming has been achieved, conventional chemotherapy can be successful, leading to better clinical outcomes for these patients.

Within this scheme, we have also included the use of OTTs as a replacement for or in combination with cisplatin, which has shown encouraging results in preclinical studies with the response of resistant tumors (Table 3). In addition, since information in these fields is still limited, it is suggested that epidrugs could potentiate the effect of immunotherapy and radiotherapy.

In line with the principles of precision medicine, the strategy is to classify tumors based on their molecular characteristics, which can guide treatment decisions to increase tumor responsiveness and improve patient quality of life while reducing side effects. However, a profound understanding of the molecular mechanisms involved in response to therapy is necessary to test this model in a clinical setting. Although significant progress has been made in describing the genetic and epigenetic features of sensitive and resistant tumors (as shown in Table 1 and Table 2), most of these findings have been based on studies conducted on cell lines that may not fully represent the complexity of the tumor microenvironment. Moreover, although important targets have been identified in these mechanisms, their effectiveness and sensitivity as biomarkers of intrinsic or acquired resistance to treatment have yet to be proven. To address these gaps in knowledge, it is imperative to conduct more preclinical and clinical trials involving large, well-characterized patient cohorts that include different population groups and with sufficient duration to describe the long-term effects of anti-cancer treatment.

## 8. Final Remarks

In TGCTs, the chemoresponse phenotype to cisplatin is defined by complex crosstalk between genetic and epigenetic factors that regulate different cellular processes, among which stand out the DNA damage response, progression of the cell cycle, and apoptosis. Seeking to restore the sensitive profile in these cells is a promising therapeutic strategy in tumors with intrinsic resistance to cisplatin and for those who need to overcome acquired resistance to chemotherapy.

It is important to continue studying the epigenetic landscape of sensitivity, which is suggested to be in itself a target that can be used to address chemoresistance through various mechanisms. In addition, epidrugs offer a promising solution to the urgent need to maintain the efficacy of cisplatin-based treatments but limit their long-term side effects. It is believed that the multimodal approach, combining chemotherapy with epidrugs and immunotherapy/radiotherapy, will be the one that produces the best results. Due to the increasing information in this context, we suggest that HDACi and DNMTi are the strongest candidates for implementation in the therapeutic management of resistant tumors in the near future. However, we are convinced that miRNAs, lncRNAs, the m6A RNA modification machinery, and chromatin remodeling complexes can also be explored as molecular targets or potential biomarkers of response to cisplatin-based therapies. Therefore, it is important to further explore these issues.

Overall, genomic and epigenetic studies, as well as their importance in chemoresponse, are still in their infancy, but their potential applications will undoubtedly shed light and promise a solution to this problem in the clinical setting.

## Figures and Tables

**Figure 1 ijms-24-07873-f001:**
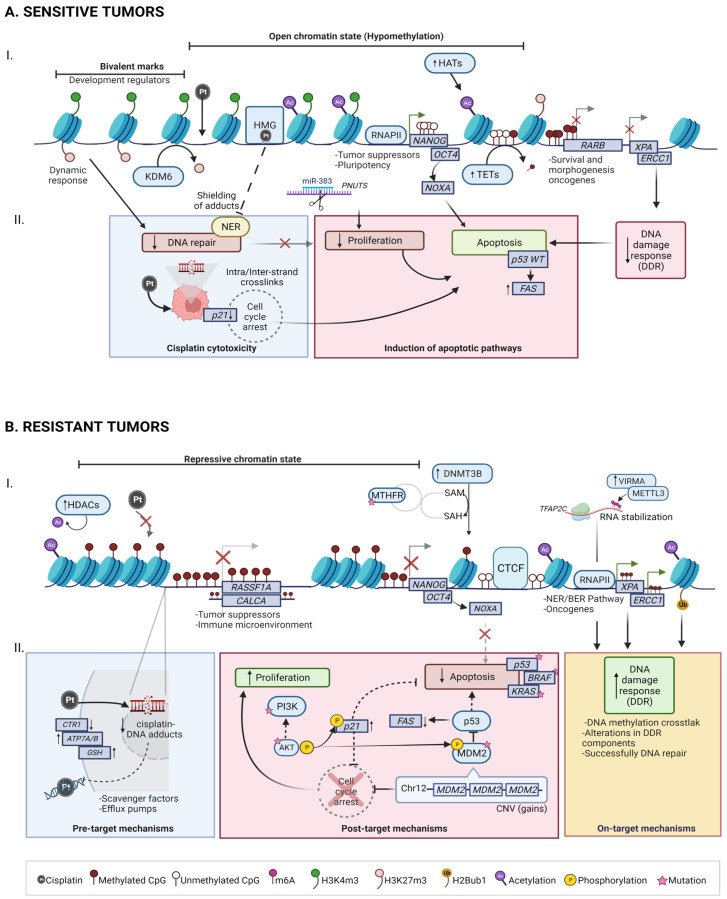
Genomic and epigenomic landscape of chemoresponse in TGCT. Proposed mechanisms that mediate sensitivity and resistance to cisplatin. (**A**) Sensitive tumors. (**AI**). Some key players that lead to cisplatin sensitivity are DNA hypomethylation, chromatin-activation marks and increased expression of pluripotency genes, as well as the crosstalk among low expression levels and promoter methylation of oncogenes. (**AII**). Intrinsic hypersensitivity is mediated by downstream regulation of DDR mechanisms induced by cisplatin, together with an open chromatin state promoting DNA–cisplatin adduct accumulation and cell cycle arrest, thus inducing cell death mediated by apoptotic pathways (mainly p53). (**B**) Resistant tumors. (**BI**). Some factors that contribute to cisplatin resistance are DNA hypermethylation, chromatin-repressive marks, increased expression of DDR factors, and decreased expression in pluripotency genes. (**BII**). Mechanisms of resistance to cisplatin are categorized into pre-target, on-target, and post-target.

**Figure 2 ijms-24-07873-f002:**
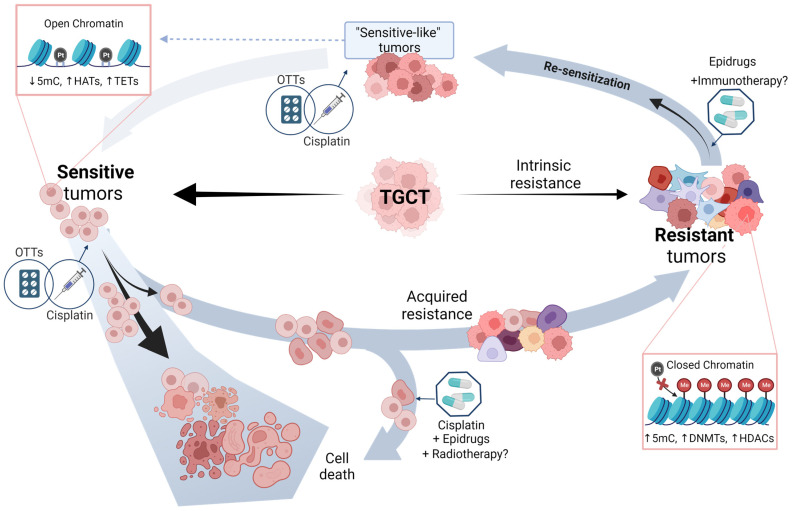
TGCT subgroups according to the phenotype of response to cisplatin. In sensitive tumors (the open-chromatin epigenetic profile), treatment with cisplatin and OTTs will preferentially lead to cell death. In tumors in which acquired resistance has developed, epidrugs administered together with cisplatin and other therapeutic strategies, such as immunotherapy and radiotherapy, will serve as a brake on progression towards the repressive/resistant epigenetic state and will potentiate the effects of the alkylating agent. On the other hand, in intrinsically resistant tumors, pre-chemotherapy epidrugs are proposed to re-sensitize cells for the subsequent administration of cisplatin. Thicker arrows distinguish the most common pathways used by tumors.

**Table 3 ijms-24-07873-t003:** Novel cisplatin-resistant TGCT treatments.

Drug Class	Therapeutic Agent	Monotherapy	Therapeutic Target/Mechanism	Study Type	Produce Relevant Response to Cisplatin-Resistant Germ Cell Tumors	Cisplatin Sensitivity-Restoring/ Improvement	Main Results	Reference
Epidrugs	Guadecitabine	Yes	HMAS	Pre-clinical	Yes	Yes	Three responses in 14 patients, with two complete responses. Inhibited progression and regressed cisplatin-resistant testicular cancer cells	[151]
	5-azacytidine	Both, combined with cisplatin	HMAS/DNMT	Pre-clinical	Yes	No	Induced apoptosis at low nanomolar doses in both cisplatin-sensitive and resistant cells	[152]
	MLo1302	Yes	HMAS/DNMT	Pre-clinical	Yes	No	Decreased cell viability by lowering the protein expression of pluripotency markers	[153]
	Decitabine	Yes	HMAS	Pre-clinical	Yes	Yes	Induced expression of tumor suppressor genes and p53 activation, encouraging a proapoptotic response and resensitizing GCT cells to cisplatin	[171,172]
	Trichostatin A/Romidepsin	Yes	HDACi	Pre-clinical	Yes	No	Antitumor activity in vitro and in vivo; induces apoptosis, reduces tumor size, and inhibits proliferation and angiogenesis	[41,173]
	Animacroxam	Yes	HDACi	Pre-clinical	Yes	No	Reduced tumor growth and angiogenesis	[154]
	Belinostat / Panobinostat	Yes	HDACis	Pre-clinical	Yes	No	Reduced acetylation, caused cell cycle arrest, decreased proliferation, lowered Ki67 index, and elevated p21, while enhancing apoptosis	[155]
	LAK-FFK11, LAK129; LAK-HGK7	Yes	Dual inhibitor (HDACi/BETi)	Pre-clinical	Yes	No	Decreased cell viability, caused apoptosis, and changed the cell cycle in cisplatin-resistant TGCT	[158]
	JQ1	Both, combined with romidepsin	BET inhibitor (BRD4)	Pre-clinical	Yes	No	Induced apoptosis, with a pronounced effect in resistant clones; reduced tumor size, proliferation rate, and angiogenesis	[157]
	C63 and BRD-K98645985	Combined with romidepsin	ARID1A (chromatin remodeler) inhibitor	Pre-clinical	Yes	Yes	Enhanced the effectiveness of romidepsin and sensitized TGCT cells to ATR inhibition	[156]
	LP99, PRT4165, GSK343, Quisinostat, JIB-04, Chaetocin and MZ-1	Yes	Epigenetic inhibitors	Pre-clinical	Yes	No	Cytotoxicity, ranging from nanomolar to micromolar. Most caused apoptosis or cell cycle arrest in GCT cell lines	[92]
	MG-132	Yes	Proteasome inhibitor	Pre-clinical	Yes	Yes	Cytotoxic in the nanomolar range for TGCT cell lines; increased sensitivity to CDDP	[159]
Inmunotherapy	BNT211	CAR T-Cell therapy	Chimeric antigen receptor	Clinical study Phase I	Yes	No	Overall response rate of 57% in a TGCT patient cohort (N=13)	[160]
Other targeted therapies	Palbociclib	Combined with cisplatin	PARP inhibitor	Pre-clinical	Yes	No	Decreased cell viability; positive effect with regard to delaying cell recovery after the insult	[167]
	Veliparib	Both, combined with cisplatin	PARP inhibitor	Pre-clinical	Yes	No	Synergistic effects when combined with cisplatin in vitro	[174]
	Olaparib	Yes	PARP inhibitor	Pre-clinical	Yes	Yes	DNA repair; sensitization to cisplatin and antitumor action	[175]
	Pazopanib	Combined with lapatinib	RTK inhibitor	Pre-clinical	Yes	No	Anti-angiogenesis properties	[41,166]
	Sunitinib	Yes	RTK inhibitor	Pre-clinical	Yes	No	In vivo antitumor action, including decreased vasculature and tumor growth inhibition	[41,165]
	Dissulfiram	Combined with cisplatin	ALDH inhibitor	Pre-clinical	Yes	Yes	An in vivo model with a synergistic antitumor effect with cisplatin	[65]
	PRI-724	Yes	Wnt/β-catenin signaling Inhibitor	Pre-clinical	Yes	No	Pro-apoptotic effects	[168]
	Metformin	Combined with cisplatin	Biguanide (antihyperglycemic agent)	Pre-clinical	Yes	Yes	Inhibited cells in the G1 phase and decreased the levels of cyclin D1, *CDK6*, *CDK4,* and *RB*; induced apoptosis	[169,170]

## Data Availability

Not applicable.

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
