# Peer review of "Breaking the Mold: Epigenetics and Genomics Approaches Addressing Novel Treatments and Chemoresponse in TGCT Patients"

_ijms, 2023, doi:10.3390/ijms24097873_

Round 1

Reviewer 1 Report

The manuscript is a review describing epigenetic and genomics features of TGCT in relation to their role in the sensitivity or resistance  to cisplatin. The topic is of interest and the it is extensively covered in this manuscript.

I have only minor suggestion:

-2.1 chapter i think that the title could be just "Histological TGCT subtypes", in order to be more consistent with following paragraphs.

- i suggest rephrase sentence in lane 145 or add more information. I don't understand who are the numerical aberrations more frequent in less aggressive tumors.

-it is not clear to me in chapter 2.2.3 when a biomarker is currently used for diagnosis or if they can be in future

- can authors indicate the six variants of KIT in lane 152?

- I found chapter 5 somehow repetitive. I suggest to merge it with chapter 4.2. Moreover if would be better to put together information about TP53 (4.1.2 - 4.2.3 - 5.1).

And Lane 438-440 and 466-472 are repetitive in previuos chapters

Author Response

We express our sincere gratitude to the reviewer for their important feedback. In response, we have made significant modifications to the manuscript (marked in yellow) in order to address each of your suggestions:

  1. The 2.1 chapter i think that the title could be just "Histological TGCT subtypes", in order to be more consistent with following paragraphs.
  2. We appreciate the suggestion and have changed the subtopic's title by "Histological TGCT subtypes" to ensure its consistency with the information presented in this review section (line 60).
  3. i suggest rephrase sentence in lane 145 or add more information. I don't understand who are the numerical aberrations more frequent in less aggressive tumors.

  1. Thanks for the suggestion, we have rephrased the sentence in line 145 as follows: "In general, the current known chromosomal aberrations have been more commonly described in less aggressive tumor subtypes" and we moved it to line 139 to make the information more consistent.

  1. it is not clear to me in chapter 2.2.3 when a biomarker is currently used for diagnosis or if they can be in future

  1. We thank the reviewer for their observation, and in order to provide accurate information, we have modified the subtopic's title (line 152) and narrative by emphasizing that the biomarkers presented are considered "potential" thus far. Furthermore, we have included additional details on the challenges of translating diagnostic genomic biomarkers to the clinic and their clinical utility (lines 155-164). Furthermore, to further clarify this, we state that although some have limited clinical use in certain hospitals, we still lack adequate information on their reproducibility and their predictive value for diagnosis.

  1. can authors indicate the six variants of KIT in lane 152?
  2. We appreciate the comment. Firstly, we have corrected the manuscript according to the cited work, and we corrected the text, starting with the fact that there are three and no six variants, we apologize for this mistake. Therefore, we have added the three KIT variants related to cryptorchidism on line 156.
  3. I found chapter 5 somehow repetitive. I suggest to merge it with chapter 4.2. Moreover if would be better to put together information about TP53 (4.1.2 - 4.2.3 - 5.1).

  1. We greatly appreciate this suggestion to our manuscript. We have merged chapter 5 with 4; basically, in lines 226-244 was positioned the general information about genomic mutations found in resistant or sensitive TGCT tumors, and in 4.1 and 4.2 sections we combine information of genomic biomarkers of chemoresponse while explaining its underlying mechanisms as can be seen in lines 317-326 for OCT4, lines 388-390 for BRAF, lines 418-423 for FGR3 and KRAS and finally, lines 408-413 for MDM2 and MYCN.

 .

As per the reviewer suggestion, we have grouped the information on chemoresponse mechanisms with biomarkers and response related to TP53 in the section on sensitivity (lines 290-302) and resistance (lines 398-404) mechanisms. We think this covers and clarify the manuscript according to the reviewer’s suggestions. We believe this is the most significant change in the manuscript and hope that the reviewer finds the information more concise after this correction.

  1. And Lane 438-440 and 466-472 are repetitive in previuos chapters

Finally, we have reviewed the lines that the reviewer found repetitive with respect to previous subtopics, as well as the entire sections, in order to remove repetitive information, according to the reviewer's pertinent suggestion.

Once again, we would like to express our gratitude to the reviewer for their insightful comments, we hope that corrections have helped to improve the quality and clarity of our manuscript.

Reviewer 2 Report

The authors provide useful potential genomic and epigenomic markers that could be useful guideline for studying the population genomics and epigenomics in personalized medicine applying to different ethnic group in the future.

Author Response

We appreciate the reviewer's comment. The manuscript was modified to make it clearer and more fluent based on the reviewers' comments. Checking that the references are the most current and eliminating repetitive regions. We add the modified version of the text for your consideration.
